# Physiological Responses of Grapevine Leaves to High Temperature at Different Senescence Periods

**DOI:** 10.3390/plants14203142

**Published:** 2025-10-12

**Authors:** Shiwei Guo, Riziwangguli Abudureheman, Zekai Zhang, Haixia Zhong, Fuchun Zhang, Xiping Wang, Mansur Nasir, Jiuyun Wu

**Affiliations:** 1College of Horticulture, Xinjiang Agricultural University, Urumqi 830052, China; 18199894023@163.com (S.G.); 13289951672@163.com (R.A.); 2Turpan Experimental Station, Xinjiang Academy of Agricultural Sciences, Xinjiang Grape Engineering Technology Research Center, Turpan 838000, China; 18992789139@163.com (Z.Z.); zhonghaixia1@sina.cn (H.Z.); zhangfc@xaas.ac.cn (F.Z.); wangxiping@nwsuaf.edu.cn (X.W.); 3Colleges of Horticulture, Northwest A&F University, Yangling 712100, China; 4Institute of Fruits and Vegetables, Xinjiang Academy of Agricultural Sciences, Urumqi 830091, China

**Keywords:** high temperature, grapevine, leaf senescence, physiological characteristics

## Abstract

Leaf senescence is a precisely regulated developmental process that is critical for grapevine growth and yield, which is easily influenced by environmental factors. High temperature is a major factor that accelerates senescence rapidly, adversely affects photosynthetic performance, severely hindering fruit nutrient metabolism and growth. This study investigated chlorophyll fluorescence and physiological traits in grape (*Vitis vinifera* L.) leaves at different senescence stages under natural high-temperature conditions in Turpan. Measurements included chlorophyll content, MDA levels, antioxidant enzyme activities, and chlorophyll fluorescence parameters. The results showed that (1) young leaves exhibited higher and more sustained chlorophyll content but were prone to wilting, whereas older leaves showed accelerated chlorosis and functional decline; (2) high temperature severely impaired PSII function, inhibiting electron transport and photochemical efficiency, reflected in increased *ABS/RC*, *TR_o_/RCC*, and *DI_o_/RC*, and decreased *F_v_/F_m_*, *F_v_/F_o_*, and *PI_abs_*; (3) POD, SOD, CAT and MDA levels initially increased then decreased, correlating with photosynthetic changes and leaf age; and (4) young leaves maintained stronger photosynthetic capability and physiological resilience than older ones. Although partial recovery occurred after temperature reduction, photosynthetic and antioxidant activities did not fully revert. This suggests persistent heat-induced functional decline and accelerated senescence, providing insights for understanding heat-induced leaf senescence and developing strategies for cultivating grapevines.

## 1. Introduction

Leaf senescence is a process of nutrient recycling from senescing leaves to nascent tissues or storage organs, crucial for plant growth, development, and environmental adaptation [1,2]. High temperature is a major stress factor inducing rapid leaf senescence, adversely affecting plant growth and development, reducing photosynthetic efficiency, and causing physiological changes such as cell membrane lipid peroxidation, severely hindering crop nutrient metabolism and growth [3]. However, most previous studies have focused on single-age leaves or controlled-environment simulations, leaving a scarcity of systematic field evidence regarding differential responses across leaf ages to prolonged extreme heat. Recent research on cucumber revealed that combined high-temperature and high-humidity stress aggravates PSII photoinhibition in older leaves, whereas young leaves maintain higher *F_v_/F_m_* and PI_abs_ [4]. A comparable age-dependent stress tolerance was observed in cotton, where melatonin-delayed drought-induced senescence was coupled with lower MDA and higher SOD/POD activities in upper-canopy leaves [5]. Chlorophyll fluorescence fingerprinting has further enabled quantification of low-temperature sensitivity in tomato yellow-leaf mutants and real-time tracking of daily senescence of Acer palmatum leaves under natural temperature fluctuations [6,7]. Transcriptome analysis in soybean showed that upper-canopy leaves activate stronger antioxidant and photosynthetic recovery pathways than mid-canopy leaves during monocarpic senescence [8]. Collectively, these studies highlight that leaf age is a key determinant of stress-induced senescence kinetics, but whether similar patterns occur in grapevines exposed to field heatwaves exceeding 40 °C remains largely unexplored.

In grapevines (*Vitis vinifera* L.), as in other crops, prolonged heat events curtail photosynthate production and berry quality by accelerating leaf senescence and impairing photosynthetic efficiency [9,10,11,12]. Turpan offers unique growing conditions for grapes due to its superior light and heat resources and diverse geography, making it a leading region for fresh and dried grape production in China. Sustained high-temperature stress, occurring on over 100 days per year above 35 °C and frequent heatwaves >40 °C, accelerates leaf senescence, damages the photosynthetic apparatus, and restricts the synthesis and translocation of photosynthetic products, thereby significantly reducing grape yield and quality [13,14]. With increasing frequency and intensity of extreme heat events due to global warming, grape-growing regions worldwide face escalating challenges from heat stress [9,15]. Therefore, this study investigated the chlorophyll fluorescence and physiological characteristics of grapevine leaves at different senescence stages under Turpan’s unique hot and arid conditions, analyzing the physiological response patterns to high temperature, to provide a reference for developing stress-resistant cultivation strategies (delaying leaf senescence).

## 2. Results

### 2.1. Temperature Dynamics in the Field

The maximum temperature recorded within the experimental plots was 47.13 °C, the minimum was 14.65 °C, and the average temperature was 32.13 °C. There were 123 days with temperatures above 35 °C, including 65 days above 40 °C (Table 1). The average temperature in May was 30.59 °C, with a maximum of 47.13 °C. There were 28 days above 35 °C, including 14 days above 40 °C. June marked the beginning of the summer high-temperature period, with a monthly average of 33.35 °C, an increase of 2.76 °C from May, and a maximum of 45.17 °C. There were 27 days above 35 °C, including 16 above 40 °C. Temperatures continued to rise in July, with a monthly average of 34.40 °C and a maximum of 44.93 °C. There were 28 days above 35 °C, including 20 above 40 °C. August experienced extreme heat, with a maximum temperature of 45.99 °C, 28 days above 35 °C, and 15 days above 40 °C. Temperatures gradually declined in September; the maximum temperature was 38.30 °C (a decrease of 7.69 °C from August’s peak), the average temperature was 26.22 °C, and the minimum dropped to 12.15 °C, bringing conditions close to the optimal temperature range for grape growth (Figure 1).

### 2.2. Phenotypic Observation of Leaves at Different Senescence Periods

Leaves at different developmental periods and high-temperature stages exhibited distinct phenotypic characteristics (Figure 2). Under natural high-temperature conditions in Turpan, grapevine leaves sequentially showed signs of senescence and phenotypic alterations induced by heat from May to September. For instance, leaves in July and August displayed noticeable wilting, chlorosis, and dried leaf margins, along with symptoms such as spotting, curling, and yellowing. Lower leaves exhibited more pronounced senescence traits like yellowing and dried edges, while symptoms of heat damage like wilting and curling were more evident in upper leaves. Middle leaves maintained relatively stable phenotypic characteristics. From May to July, leaves at all positions (upper, middle, and lower) appeared fresh green to dark green with fully expanded shapes. In July, lower leaves began showing marginal scorching and partial chlorosis. Subsequently, middle and lower leaves also exhibited marginal scorching and chlorosis in August, while upper leaves remained green but developed yellow spots, indicating that high temperature might accelerate the senescence process. By September, lower leaves showed significant withering, yellowing with spots, the yellowing area in middle leaves expanded further with curled and dehydrated margins, whereas upper leaves remained relatively healthy and intact, with a dark green color. This suggests high temperature may cause chlorophyll degradation or inhibit synthesis, accelerating leaf senescence and yellowing, particularly in lower leaves, which senesced over a month earlier than upper leaves. In summary, young leaves are prone to water loss and wilting under high temperature induction, whereas older leaves tend towards chlorosis and senescence, consistent with plant leaf growth patterns and representing a self-protection mechanism in response to heat stress.

### 2.3. Chlorophyll Content of Grapevine Leaves at Different Senescence Stages

Chlorophyll, the primary pigment for photosynthesis in grapevine leaves, directly reflects their nutritional status and photosynthetic capacity. High temperature inhibits chlorophyll synthesis and accelerates its degradation, forcing leaves into functional decline. The results showed that under prolonged natural high temperature stress, the chlorophyll content in grapevine leaves from different periods exhibited a dynamic trend of initial increase followed by a decrease (Figure 3a). Chlorophyll content peaked in June (34.72 SPAD), which was significantly higher than that in August and September, and reached its lowest value in September (31.53 SPAD). However, significant differences existed among leaf positions. From 10 May to 6 August, chlorophyll content previously showed lower leaves > middle leaves > upper leaves. The initial chlorophyll content was highest in lower leaves (36.93 SPAD), followed by middle leaves (31.61 SPAD), and lowest in upper leaves (25.02 SPAD). Starting from 14 August, the chlorophyll content of lower leaves was gradually surpassed by middle leaves, but remained higher than upper leaves. By 30 September, the order of chlorophyll content became middle leaves (33.04 SPAD) > upper leaves (31.55 SPAD) > lower leaves (29.66 SPAD).

These results indicate that the chlorophyll content of lower leaves continuously decreased with rising temperature throughout the experiment, accompanied by gradual chlorosis and senescence (Figure 2), while middle and upper leaves maintained relatively higher levels (Figure 3b). The chlorophyll content of upper leaves remained stable and even increased slightly during the mid-to-late high temperature period (11 July), albeit with a small increase never exceeding the maximum value of lower leaves, possibly related to high temperature affecting its chlorophyll synthesis. Overall, younger leaves (upper and middle) exhibited higher and more sustained chlorophyll content compared to older leaves (lower) during the high-temperature period. This pattern aligns with natural leaf development and senescence dynamics and is consistent with the temperature variations recorded in the experimental plots (Figure 1 and Figure 3).

### 2.4. Effects of High Temperature on the OJIP Curve of Leaves at Different Senescence Stages

Standardized curves for different leaf positions and periods showed that all parameter curves exhibited the typical O-J-I-P phase transition characteristics. Fluorescence intensity increased in an “S” shape over time, eventually reaching a steady state (P point), consistent with the response of plant photosystem II (PSII) to light excitation (Figure 4). However, significant differences existed in fluorescence intensity and phase transition slopes among leaves at different senescence stages. Firstly, the OJIP curves overall showed a gradual increase from May to August, followed by a slight decrease in September, reflecting possible damage to PSII reaction centers during the summer high temperature period and subsequent initiation of repair mechanisms. The overall trend was September > August > July > June > May (Figure 4f). Secondly, as leaves grew and temperature gradually increased, the OJIP curves of upper leaves changed, with the OJ phase gradually being surpassed by lower and middle leaves, yet photosynthetic capacity remained relatively higher than lower leaves (August, September). Specifically, the OJ phase continued to rise, more noticeably in middle and lower leaves than upper leaves. The efficiency of Q_A_ to Q_B_ electron transfer fluctuated with temperature, with the degree of impediment following August > July > June > September > May. By September, the inhibition extent ranked as lower leaves > middle leaves > upper leaves, indicating differential heat-induced damage to PSII among leaf positions, which aligns with observed leaf senescence phenotypes.

Fluorescence intensity increased with leaf growth and rising temperature within the entire experimental period, and the shape of the OJIP curve changed accordingly. The O point (20μs) reflects the open state of PSII, the J point (2 ms) indicates Q_A_ accumulation and electron transfer efficiency, the K point reflects the activity of the oxygen-evolving complex (OEC), and the I-P phase (30–300 ms) relates to the thylakoid proton gradient. Double-normalized OJIP curves revealed a bimodal curve, with L, K, J, and I points gradually shifting upward (Figure 4g). However, the P point (maximum fluorescence intensity *F_v_/F_m_*) was significantly lower than in May and showed a decreasing trend, indicating that rising temperature led to a gradual decrease in *F_v_/F_m_* (Figure 4g), with some variation among leaf positions. In the early stage (May, June, and July), the order was generally upper leaves > middle leaves > lower leaves (Figure 4a,b), with photosynthetic capacity decreasing accordingly. As the temperature increased, leaf fluorescence intensity and damage to PSII reaction centers intensified, with the K-J phase change being most significant. Lower leaves showed a more pronounced increase than middle and upper leaves, suggesting OEC deactivation due to high temperature, with significant variation among leaf positions. The P phase reflects the steady-state fluorescence level of PSII reaction centers under continuous light. The overall amplitude decreased by 6.3% from May to July and increased by 2.6% from August to September. The time required to reach the P phase was about 800 ms in May, nearly 950 ms in July (+18.8%), and recovered to 820 ms in September, indicating that high temperature affected the photosynthetic system, with some recovery as temperature decreased, but significant differences existed among leaf positions, with younger leaves showing more pronounced changes.

### 2.5. Chlorophyll Fluorescence Parameters of Grapevine Leaves at Different Senescence Stages

*F_v_/F_m_* represents the maximum photochemical efficiency, *F_v_/F_o_* indicates potential photochemical activity, *PI_abs_* comprehensively assesses the photosynthetic performance of PSII, *φ*(*E_o_*) reflects the actual photochemical efficiency of PSII, *φ*(*R_o_*) characterizes the actual photochemical efficiency of PSI, and *ψ*(*E_o_*) represents electron transport efficiency. Analysis of chlorophyll fluorescence parameters (Figure 5a) showed that these parameters overall exhibited similar patterns, decreasing with prolonged high temperature exposure. *F_v_/F_m_*, *F_v_/F_o_*, *PI_abs_*, *φ*(*E_o_*), and *φ*(*R_o_*) all peaked in May, reached their lowest points in August, and recovered somewhat by September, indicating that high temperature inhibited the activity and efficiency of PSII reaction centers, with longer stress duration leading to more pronounced inhibition. *ψ*(*E_o_*), reflecting electron transport efficiency and proportional to photosynthetic performance parameters, was highest in June, lowest in August, and recovered slightly in September, suggesting high temperature limited electron transport, with longer exposure having a more significant impact. *F_o_/F_m_* reflects potential damage to PSII reaction centers, *δ*(*R_o_*) assesses the redox balance state of the photosynthetic apparatus, *V_j_* reflects the efficiency of electron transport on the acceptor side of PSII (particularly from Q_A_ to Q_B_), and *V_i_* is related to the oxidation state of the donor side OEC and the PQ pool. Results showed *F_o_/F_m_* first increased and then decreased; *δ*(*R_o_*) decreased initially, then increased; and *V_j_* values followed August > July > June > September > May, where higher *V_j_* values indicate more impeded electron transport. These findings suggest that high temperature caused severe damage to both the acceptor and donor sides of PSII, leading to a significant imbalance between carbon assimilation and photorespiration (Figure 5b).

Analysis of energy parameters per cross-section (*ABS/CS_o_*, *DI_o_/CS_o_*, *TR_o_/CS_o_*, *ET_o_/CS_o_*, *RE_o_/CS_o_*, and *TR_o_/CS_o_*) revealed that from May to August, *ABS/CS_o_* and *DI_o_/CS_o_* increased with rising temperature and decreased with falling temperature by September, indicating that high temperature increased light energy absorption and heat dissipation per unit leaf area (Figure 5c). Parameters including *TR_o_/CS_o_*, *ET_o_/CS_o_*, and *RE_o_/CS_o_* showed an initial increase followed by a decrease, with *TR_o_/CS_o_* reaching its peak in August, suggesting that high temperature affected energy distribution and the energy available for reducing Q_A_, related to the duration of high temperature. Evaluation of energy flux parameters per reaction center (*ABS/RC*, *DI_o_/RC*, *RE_o_/RC*, *TR_o_/RC*, and *ET_o_/RC*) showed that *ABS/RC*, *DI_o_/RC*, and *TR_o_/RC* exhibited an inverted V-shaped pattern, seen in Figure 5d, increasing from May to August and decreasing by September. *ET_o_/RC* (energy used for electron transport per RC) peaked in June and subsequently decreased continuously, while *RE_o_/RC* (energy transported to the end of the electron chain per RC) decreased gradually from May to July and then increased again. These indicate that high temperature significantly affected electron transport from absorbed light energy in the reaction centers, leading to increased absorption and thermal dissipation of light energy, with longer exposure having a more pronounced effect.

### 2.6. Effects of High Temperature on Antioxidant Enzyme Activities and MDA Content

The activities of antioxidant enzymes (POD, SOD, and CAT) in grapevine leaves from May to September all exhibited a trend of initial increase followed by a decrease (Figure 6a–c), with peak levels observed in August that were significantly higher than the initial values in May. Although a decline was detected in September, the enzyme activities remained significantly elevated compared to those in May. A similar variation pattern was observed for MDA content (Figure 6d), which was lowest in May (15.16 nmol/g DW) and reached its maximum in August (37.87 nmol/g DW). During the high-temperature period (June to August), MDA content increased continuously, showing increments of 33.51%, 85.09%, and 149.74% compared to the initial value, respectively. The highest MDA content was recorded in August (37.87 nmol/g DW), representing the most pronounced increase. By September, MDA content had decreased to 26.39 nmol/g DW, a reduction of 30.32% compared to August, yet it remained significantly higher than the initial level in May, with a recovery rate of 74.02%. These results indicate that POD, SOD, CAT, and MDA levels change synchronously in response to temperature variations, showing a coordinated initial rise and subsequent decline. Furthermore, the upper leaves exhibited relatively higher antioxidant enzyme activities and lower MDA content compared to the middle and lower leaves, suggesting that younger leaves possess a stronger physiological responsiveness to heat stress than older leaves.

### 2.7. Correlation Analysis of Various Indicators

Correlation analysis of chlorophyll content, chlorophyll fluorescence parameters, antioxidant enzyme activities, and MDA content in grapevine leaves at different senescence stages under high temperature conditions revealed certain correlations among all indicators (Figure 7). *F_o_* was significantly positively correlated with *F_m_* (0.921 *), SOD (0.956 *), and CAT (0.943 *) (*p* < 0.05), and highly significantly positively correlated (*p* < 0.01) with *F_o_/F_m_* (0.983 **), *ABS/RC* (0.985 **), *DI_o_/RC* (0.971 **), *TR_o_/RC* (0.982 **), *ABS/CS_o_* (1.000 **), *DI_o_/CS_o_* (0.986 **), *TR_o_/CS_o_* (0.988 **), and MDA (0.993 **). However, *F_o_* was significantly negatively correlated with *φ*(*R_o_*) (−0.918 *) and *PIabs*(−0.881 *) (*p* < 0.05) and highly significantly negatively correlated (*p* < 0.01) with *F_v_/F_m_* (0.984 **) and *F_v_/F_o_* (−0.987 **). *F_m_* was significantly positively correlated (*p* < 0.05) with *F_o_*, *TR_o_/RC*, *ABS/CS_o_*, *POD*, *SOD*, and *CAT*, and highly significantly correlated with *V_i_* (0.968 **) and *TR_o_/CS_o_* (0.969 **). Simultaneously, *F_m_* showed a significant negative correlation (*p* < 0.01) with *φ*(*R_o_*) and *F_v_/F_o_*, with correlation coefficients of −0.993 ** and −0.900 *, respectively. *F_v_/F_m_* was significantly positively correlated with *F_v_/F_o_* (0.994 **) and *PI_abs_* (0.929 **) and highly significantly negatively correlated (*p < 0.01*) with *F_o_*, *F_o_/F_m_*, *ABS/RC*, *DI_o_/RC*, *TR_o_/RC*, *ABS/CS_o_*, *DI_o_/CS_o_*, and MDA. It was also negatively correlated (*p < 0.05*) with *TR_o_/CS_o_* (−0.956 *) and SOD (−0.899 *)*. V_j_* was highly significantly negatively correlated with *ψ*(*E_o_*) and *φ*(*E_o_*) (*p* < 0.01; coefficients −1.000 ** and −0.969 **). *V_i_* was highly significantly negatively correlated (*p* < 0.01) with *RE_o_/RC* and *φ*(*R_o_*) (coefficients −0.961 ** and −0.988 **).

Within the energy allocation module, parameters at the reaction center level, like *ABS/RC*, *TR_o_/RC*, and *DI_o_/RC*, formed distinct clusters. *ABS/RC* was highly significantly positively correlated (*p* < 0.01) with *DI_o_/RC*, *TR_o_/RC*, *ABS/CS_o_*, *DI_o_/CS_o_*, and MDA, and significantly negatively correlated with *PI_abs_* (*p* < 0.05). *ψ*(*E_o_*) was highly significantly positively correlated with *φ*(*E_o_*) (*p* < 0.01, coefficient 0.969 **). *φ*(*R_o_*) was significantly negatively correlated with *A ABS/CS_o_*(−0.918 *), *TR_o_/CS_o_*(−0.961 **), POD(−0.959 **), SOD(−0.928 *), and CAT (−0.966 **) (*p* < 0.05 for *, *p* < *0.01* for **). Chlorophyll (*Chl*) content was only significantly negatively correlated with *RE_o_/CS_o_* (−0.920 *). Oxidative stress module: POD was significantly positively correlated with SOD (0.928 *) and CAT (0.980 *), while SOD and CAT were also highly significantly correlated (0.981 *, *p* < 0.05). MDA was significantly positively correlated with SOD (0.925 *) and CAT (0.900 *) and highly significantly negatively correlated (*p < 0.01*) with photosynthetic efficiency parameters like *F_v_/F_m_*, *F_v_/F_o_*, and *PI_abs_*, as well as with parameters indicative of photodamage and thermal dissipation such as *F_o_*, *F_o_/F_m_*, *ABS/RC*, *DI_o_/RC*, *TR_o_/RC*, *ABS/CS_o_*, *DI_o_/CS_o_*, and *TR_o_/CS_o_*.

## 3. Discussion

High-temperature-induced functional leaf senescence is a major factor limiting crop yield and quality [15]. With rising temperatures due to global warming, high temperatures have become a primary stress factor constraining grape production and quality. Leaves, as a crucial location for photosynthesis and gas exchange, exhibit morphological changes that serve as prominent indicators of heat stress in grapevines. These changes not only determine leaf structure and function but also directly reflect the extent of heat damage and senescence [16,17]. This study found that under natural high-temperature conditions in Turpan, young leaves are prone to water loss and wilting induced by heat, while older leaves are more susceptible to chlorosis and senescence. Chlorophyll, the primary pigment for photosynthesis, changes in content are a key feature of leaf senescence [18]. High temperature inhibits chlorophyll synthesis and accelerates its degradation, forcing leaves into functional decline and impairing photosynthesis [19,20,21,22]. This study found chlorophyll content significantly decreased with sustained temperature increase, consistent with findings in grapevines [23], tomatoes [24], maize [25], and other crops. However, chlorophyll content in young leaves was relatively higher and persisted longer than in old leaves. Furthermore, chlorophyll content in old leaves continued to decline even after temperatures decreased, indicating irreversible damage and rapid senescence abscission in these functional leaves, potentially a self-protection mechanism in response to heat stress.

Photosynthesis is one of the most heat-sensitive processes in plants. Damage to any component of the photosynthetic apparatus—photosynthetic pigments, Photosystem I (PSI), Photosystem II (PSII), electron transport chain, and CO_2_ release pathways—can hinder photosynthesis [26,27,28]. PSII is particularly sensitive. This study found that sustained high temperature severely damaged both donor and acceptor sides of PSII. Donor side damage was manifested by a significant rise in the K phase of the OJIP curve under high temperature, inhibiting photosynthetic electron transfer and photochemical reactions, reducing the conversion efficiency of light energy in PSII reaction centers, consistent with previous reports [29,30]. Acceptor side damage was shown by a significant increase in fluorescence intensity at the J point, indicating impeded electron transfer from *Q_A_* to *Q_B_* and increased heat dissipation [31]. Significant increases in *ABS/RC*,* TR_o_/RC*,* DI_o_/RC*, and a significant decrease in *ET_o_/RC* suggested that ATP synthesized by photochemical reactions in the early stage of high temperature could still be used for the Calvin cycle to prevent damage from over-excitation in the PSII antenna system. However, as heat stress duration increased, energy transfer between the light-harvesting complex (LHCII) and PSII was inhibited, causing significant decreases in key photosynthetic efficiency parameters like *F_v_/F_m_*, *F_v_/F_o_*, and *PI_abs_* [32,33,34,35], consistent with results in rice [36], cucumber [37], rhododendron [38], etc. Additionally, the authors found photosynthetic efficiency parameters recovered slightly as temperatures decreased later, with recovery being more significant in younger leaves, but not fully to initial levels. This is likely related to functional decline in grapevine leaves caused by high temperature.

High temperature leads to massive ROS accumulation and membrane lipid peroxidation. MDA, a key product of peroxidation, can further exacerbate membrane lipid peroxidation, causing dysfunction of cell membranes and macromolecules, and physiological metabolic imbalance [39]. This study found MDA content changed synchronously with temperature, with higher temperatures correlating with higher MDA content. To scavenge excess ROS, plants increase the activities of POD, CAT, and SOD to reduce damage to the cytoplasmic membrane and maintain normal cellular function. Antioxidant enzyme activities peaked during extreme heat and decreased as temperatures fell, indicating a time-dependent physiological response to heat stress [40,41], with variations among leaf ages. For instance, upper leaves had relatively higher antioxidant enzyme activities and lower MDA content compared to middle and lower leaves, suggesting young leaves possess stronger physiological responsiveness than old leaves [2,42].

## 4. Materials and Methods

### 4.1. Description of the Study Area

The experiment was conducted in the grapevine germplasm orchard of the Turpan Institute of Agricultural Sciences, Xinjiang Academy of Agricultural Sciences (XAAS). The garden is located at 89°11′ E, 42°56′ N, at an altitude of 0 m, which is located in Gaochang district, Turpan city, Xinjiang, characterized by an extremely hot and arid environment. The annual average temperature is 16.3 °C, with more than 100 days exceeding 35 °C. The summer extreme maximum temperature reaches 52.2 °C. The annual average precipitation is only 16.0 mm, while the annual evaporation exceeds 3600 mm, annual sunshine duration is 2788.4 h in this area, which was classified as a typical continental warm-temperate desert climate.

### 4.2. Plant Materials

Two-year-old (2a) ‘Thompson Seedless’ (*V. vinifera*) grapevine plants with almost the same growth were selected as experimental materials. Seedlings were field-cultivated using a pergola system with a ‘V’-shaped canopy, planted in 1.5 m × 1.0 m plots with north–south row orientation. Furrow irrigation was applied, which was typically performed every 7 to 15 days based on air temperature and soil moisture levels. The soil was classified as a sandy loam. The grapevine seedlings demonstrated moderate growth vigor and were managed according to conventional horticultural practices.

### 4.3. Measurements and Methods

#### 4.3.1. Temperature Monitoring

The experiment was conducted from 1 May to 30 September in 2024. Temperature was monitored at fixed points in the grapevine garden of the Turpan Agricultural Science Research Institute, XAAS. We employed the MicroLite USB temperature data record device (manufactured by Fourier Systems, Fourtec-Fourier Technologies, Ltd., San Francisco, CA, USA) to monitor temperatures at a specific location. Measurements were recorded hourly throughout the entire experimental period (May to September).

#### 4.3.2. Chlorophyll Content Measurement

Representative upper, middle, and lower leaves were selected and marked sequentially, constituting approximately half of the total leaf population. During the natural high-temperature period in Turpan, marked leaves were measured at scheduled times. The relative chlorophyll content of grapevine leaves was determined using a SPAD-502 Plus chlorophyll meter (Konica Minolta, Chiyoda-ku, Tokyo, Japan), expressed as SPAD values. Measurements were taken approximately every 7–10 days, with 3 leaves per treatment and 3 biological replicates, between 11:00 and 13:00. Phenotypic characteristics of leaves were recorded each time as well.

#### 4.3.3. Chlorophyll Fluorescence Parameters

During the entire experimental period with the natural high-temperature period in Turpan, the fully opened grapevine leaves from different positions, including upper, middle, and lower leaves, were selected and marked sequentially, accounting for almost half of the whole plant’s leaves. On clear, windless days, photosynthetic characteristics of marked leaves were measured at scheduled times, approximately every 7–10 days, using the OJIP test. Chlorophyll fluorescence parameters (*F_o_*, *F_m_*, *F_v_*, *F_v_/F_o_*, *F_v_/F_m_*, *V_j_*, *V_i_*, *ABS/RC*, *DI_o_/RC*, *TR_o_/RC*, *ET_o_/RC*, *RE_o_/RC*, *ψ*(*E_o_*), *φ*(*E_o_*), *δ*(*R_o_*), *φ*(*R_o_*), *ABS/CS_o_*, *DI_o_/CS_o_*, *TR_o_/CS_o_*, *ET_o_/CS_o_*, *RE_o_/CS_o_*, *PI_abs_,* etc.) were measured by Handy-PEA fluorometer (Hansatech Instruments Ltd., Norfolk, England, UK) to provide information on the photochemical activity of PSII and the status of the PQ-pool. The sun-exposed grapevine leaves were dark-adapted for 20 min using leaf clips before measurement, and fluorescence was induced by saturating light at 3500 μmol/(m^2^·s).

#### 4.3.4. Antioxidant Enzyme Activities and MDA Content

Sampling occurred between 11:00 and 13:00 during each period. Functional leaves from upper, middle, and lower positions with similar orientation and growth vigor were selected, placed in prepared sealed bags, immediately snap-frozen in liquid nitrogen, transported to the laboratory, and stored at −80 °C. Samples were then lyophilized to constant weight, and the dry mass (DW) was recorded; all subsequent calculations were performed on a DW basis. 0.1 g DW of grapevine leaf tissue was weighed and placed into a mortar (pre-cooled on ice). A small amount of quartz sand was added along with 1.0 mL of prepared extraction buffer, and the tissue was ground into a homogenate on ice. The homogenate was centrifuged at 12,000× *g* for 10 min at 4 °C, and the supernatant (enzyme extract) was transferred to a 2.0 mL centrifuge tube and stored at 0–4 °C for assay. Superoxide dismutase (SOD, M0102B), peroxidase (POD, M0105B), catalase (CAT, M0103B), and malondialdehyde (MDA, M0106B) were quantified using respective colorimetric assay kits (Suzhou Mengxi Biomedical Technology Co., Ltd., Suzhou, China) according to the manufacturer’s instructions. SOD and POD activities are expressed as U/g DW, CAT activity as nmol/min/g DW, and MDA concentration as nmol/g DW.

### 4.4. Data Analysis

The test data were used for variance analysis (ANOVA), employing the LSD method for multiple comparisons and assessing the significance of differences. The level of *p < 0.05* was considered statistically significant, while *p <* 0.01 denoted a highly significant distinction. Correlation analysis and ANOVA, and figure generation were conducted using GraphPad Prism ver. 10.6.0 (796) (Dotmatics Corporation, Boston, MA, USA), OriginPro 2025 ver. 10.2.0.188 (OriginLab Corporation, Northampton, MA, USA) and IBM SPSS Statistics ver. 26 (June 2019, IBM Corporation, Armonk, NY, USA).

## 5. Conclusions

High temperatures induce functional senescence in grapevine leaves in an age-dependent manner. Compared to older leaves, younger leaves exhibit enhanced thermotolerance, characterized by delayed chlorophyll degradation, reduced MDA accumulation, and elevated activities of antioxidant enzymes such as POD, CAT, and SOD. Photosystem II (PSII) is particularly susceptible to heat stress, as evidenced by perturbations in OJIP transient kinetics and a decline in photosynthetic parameters, including *F_v_/F_m_* and *PI_abs_*. Although partial recovery of photosynthetic function is observed following temperature reduction, such recovery is incomplete and age-dependent, with older leaves sustaining irreversible damage. It is noteworthy that leaf senescence under high-temperature stress represents not only a passive consequence of physiological dysfunction but also an active adaptive strategy—possibly a self-protective mechanism involving complex regulatory pathways that require further in-depth study.

## Figures and Tables

**Figure 1 plants-14-03142-f001:**
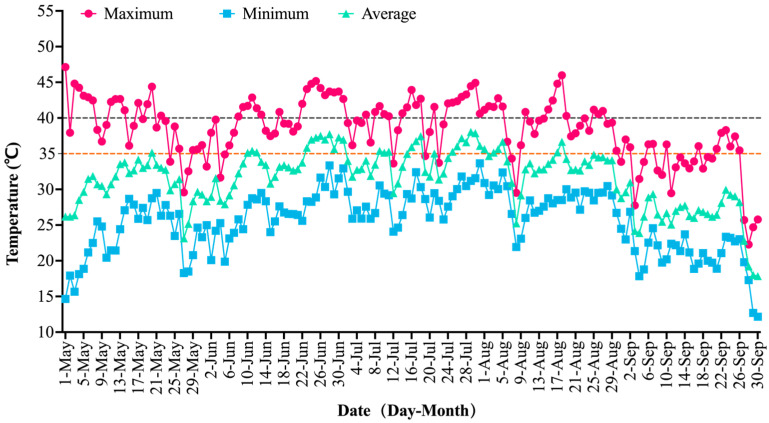
Daily temperature dynamics in the Turpan grape-growing region during the high-temperature period in 2024. Horizontal dashed lines at 35 °C and 40 °C indicate the temperature thresholds used in Table 1 for counting days with daily maximum ≥35 °C and ≥40 °C, respectively.

**Figure 2 plants-14-03142-f002:**
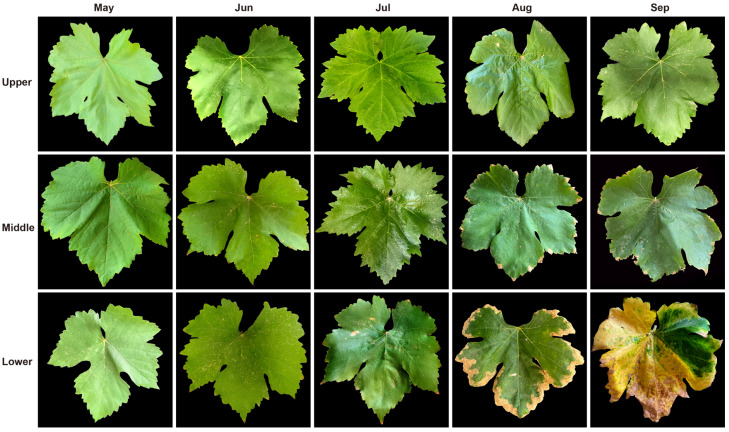
Phenotypic characteristics of grapevine leaves at different senescence stages under natural high-temperature conditions in Turpan. (**Upper**), (**middle**), and (**lower**) leaves are shown from May to September.

**Figure 3 plants-14-03142-f003:**
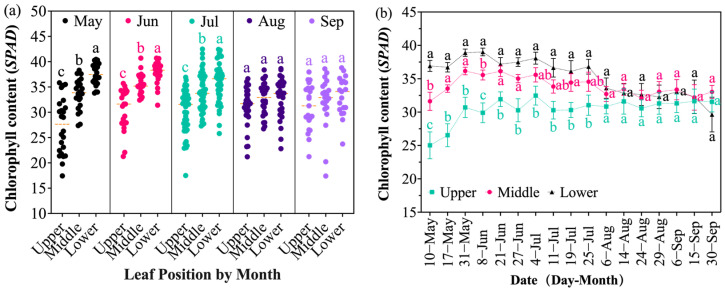
Changes in chlorophyll content of grapevine leaves at different senescence stages: (**a**) Monthly changes in chlorophyll content by leaf position (upper, middle, and lower); different lowercase letters indicate significant differences (*p* < 0.05) among leaf positions within the same month. (**b**) Time-course changes in chlorophyll content by leaf position; different lowercase letters indicate significant differences (*p* < 0.05) among leaf positions at the same sampling date.

**Figure 4 plants-14-03142-f004:**
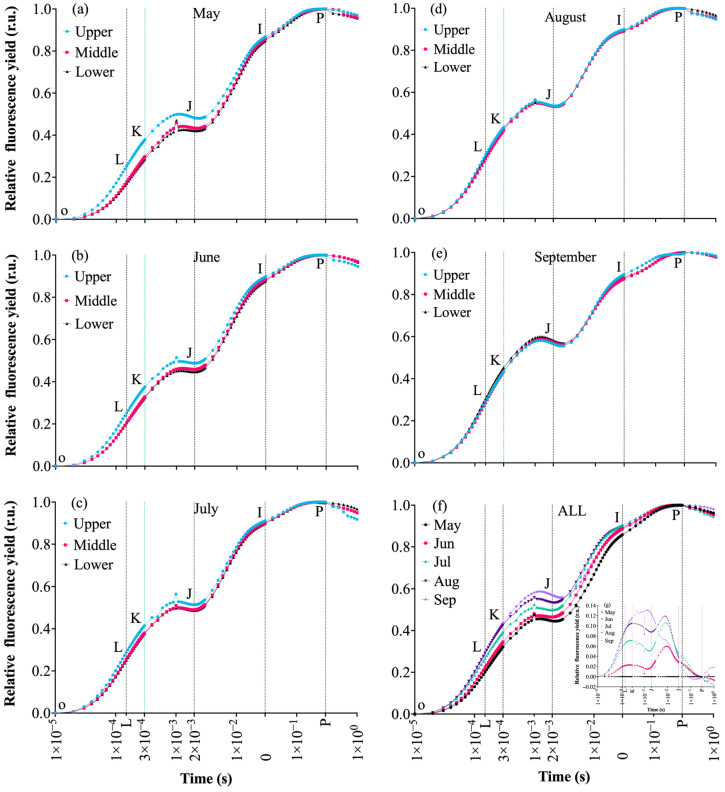
Standardized OJIP curves of grapevine leaves at different senescence stages: (**a**–**e**) OJIP curves for upper, middle, and lower leaves in May, June, July, August, and September, respectively. (**f**) Monthly comparison of OJIP curves (inset: (**g**), double-normalized OJIP curves). Vertical dashed lines mark the standard OJIP phases.

**Figure 5 plants-14-03142-f005:**
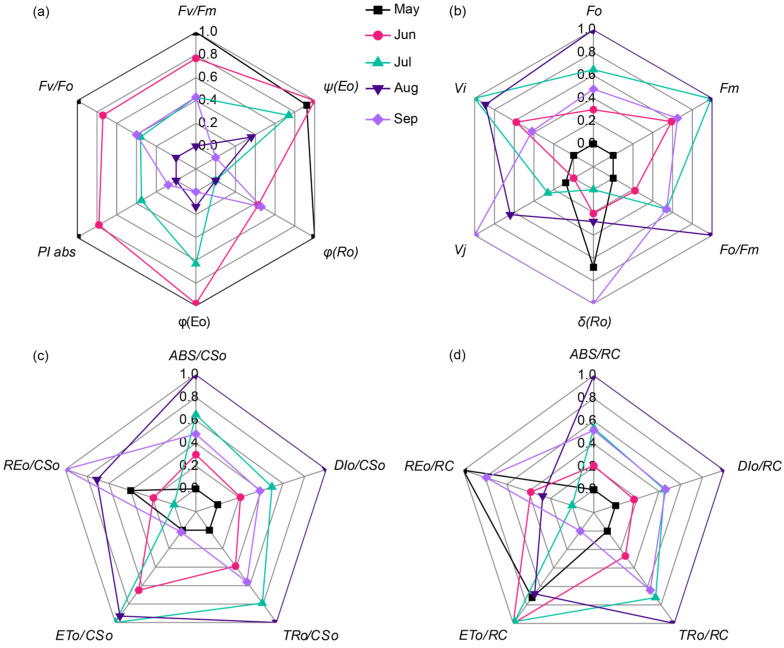
Radar plots of chlorophyll fluorescence parameters in grapevine leaves at different senescence stages: (**a**) photosynthetic efficiency parameters: *F_v_/F_m_*, *F_v_/F_o_*, *PI_abs_*, *φ*(*E_o_*), *ψ*(*E_o_*), and *φ*(*R_o_*); (**b**) PSII structural and functional parameters: *Fo*, *V_i_*, *V_j_*, *δ*(*R_o_*), *F_o_/F_m_*, and *F_m_*; (**c**) energy flux parameters per cross-section: *ABS/CS_o_*, *RE_o_/CS_o_*, *ET_o_/CS_o_*, *TR_o_/CS_o_*, and *DI_o_/CS_o_*; (**d**) energy flux parameters per reaction center: *ABS/RC*, *RE_o_/RC*, *ET_o_/RC*, *TR_o_/RC*, and *DI_o_/RC*.

**Figure 6 plants-14-03142-f006:**
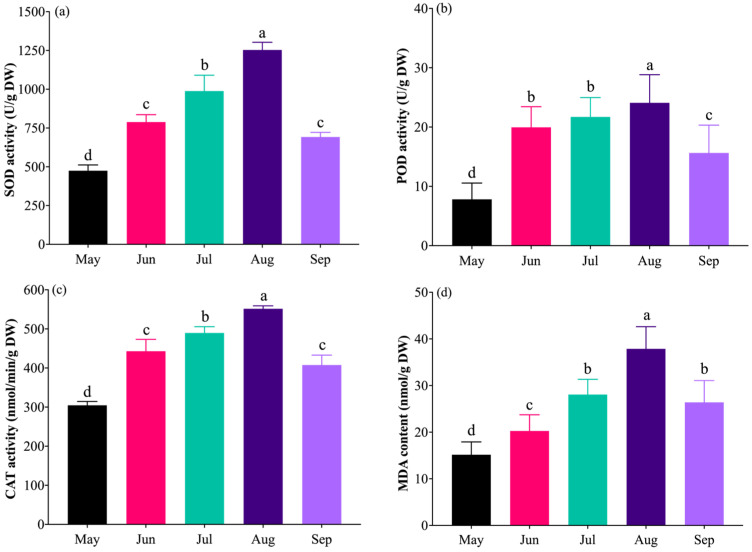
Antioxidant enzyme activity and MDA content in grapevine leaves at different stages of the high-temperature period: (**a**) monthly SOD activity; (**b**) POD activity; (**c**) CAT activity; (**d**) MDA concentration. All data are expressed on a liquid-nitrogen-quenched, lyophilized dry mass (DW) basis: SOD and POD in U/g DW, CAT in nmol/min/g DW, MDA in nmol/g DW. Units: SOD unit is defined as the amount of enzyme causing 50% inhibition of NBT reduction per minute and expressed as U/g DW. POD unit is defined as a 0.01 increase in A_470_ per minute per gram of dry tissue. The CAT unit is defined as 1 nmol H_2_O_2_ decomposed per minute per gram of dry tissue. Different lowercase letters above the bars indicate significant differences (*p* < 0.05) among months.

**Figure 7 plants-14-03142-f007:**
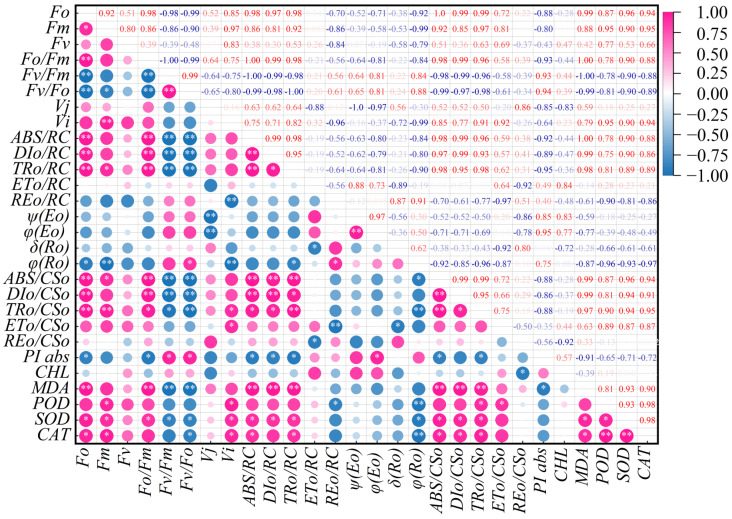
Correlation matrix of physiological and biochemical indicators in grapevine leaves under high-temperature stress. ** and * indicate significant correlations at *p <* 0.01 and *p <* 0.05, respectively.

**Table 1 plants-14-03142-t001:** Temperature variation during the high temperature period (2024).

Temperature	May	Jun	Jul	Aug	Sep
Maximum Temperature/°C	47.13	45.17	44.93	45.99	38.30
Minimum Temperature/°C	14.65	19.87	24.08	21.93	12.15
Average Temperature/°C	30.59	33.35	34.4	33.27	26.22
≥35.00 °C/Day	28	27	28	28	12
≥40.00 °C/Day	14	16	20	15	0

Note: The maximum temperature refers to the average of the daily maximum temperature in that of month, the minimum temperature refers to the average of the minimum temperature in that of month, the average temperature refers to the average of the daily average temperature in that of month, and the duration of ≥35 °C and ≥40 °C refers to the time when the average daily maximum temperature of the month is greater than ≥35 °C and ≥40 °C.

## Data Availability

The data that support the findings of this study are available upon request from the corresponding author.

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
