# Peer review of "Physiological Responses of Grapevine Leaves to High Temperature at Different Senescence Periods"

_plants, 2025, doi:10.3390/plants14203142_

Round 1

Reviewer 1 Report

Comments and Suggestions for Authors

Review of the article by Guo Shiwei et al.: »Physiological Responses of Grapevine Leaves to High Temperature at Different Senescence Periods»

 In this work, the authors investigated chlorophyll fluorescence and the physiological characteristics of grape leaves at different stages of aging under natural high-temperature growing conditions. It was shown that the young leaves retained a stronger photosynthetic ability and physiological stability than the older ones. A fairly good analysis and discussion of the photochemical activity of young leaves, middle-tier leaves and old leaves is given.

The authors conclude that the results obtained make it possible to better understand the aging of leaves caused by high temperature and develop strategies for growing grapes. We can agree with this, but the article would benefit significantly if the authors gave the percentage distribution of leaves per plant during the growing season and their contribution to the overall photosynthetic productivity of plants.

Nevertheless, the results are of interest to bioecologists and may be published in a journal.

Author Response

Q.1. The article would benefit significantly if the authors gave the percentage distribution of leaves per plant during the growing season and their contribution to the overall photosynthetic productivity of plants.

Answer: We totally agree with your suggestions. We have added the detailed information about the percentage distribution of leaves per plant in 4.3 Measurements and Methods. (Line 580, 594-595).

Reviewer 2 Report

Comments and Suggestions for Authors

In my opinion, the article cannot be accepted in its present form, because the results in Figures are not presented in a reader-friendly manner. The figure captions are too short and uninformative. For example, in Figure 3, there is no explanation for the meanings of columns, bars, and points. Even the difference between a and b parts of the figure is not explained. In b part, the results seem to differentiate between upper, middle, and lower (leaves?), but any explanation about statistical differences between different leaf categories is lacking. Meanwhile, Figure 3a lacks these categories altogether, and it is not explained why. Besides lacking informative captions, some of the figures are simply too small, especially, Figure 4. The contents of the figure are barely visible in the pdf format.

Since the statistical differences between different leaf categories are not shown or explained, it remains unclear whether the claims of the authors, regarding the differences between "younger" and "older" leaves can be justified.

Author Response

Q.1. The results in Figures are not presented in a reader-friendly manner. The figure captions are too short and uninformative. For example, in Figure 3, there is no explanation for the meanings of columns, bars, and points. Even the difference between the a and b parts of the figure is not explained. In the b part, the results seem to differentiate between upper, middle, and lower (leaves?), but any explanation about statistical differences between different leaf categories is lacking. Meanwhile, Figure 3a lacks these categories altogether, and it is not explained why.

Answer: Thank you so much for your detailed suggestions. We have provided more information about each figure and revised the manuscript as per the recommendations.

Q.2. Besides lacking informative captions, some of the figures are simply too small, especially, Figure 4. The contents of the figure are barely visible in the PDF format. Since the statistical differences between different leaf categories are not shown or explained, it remains unclear whether the claims of the authors, regarding the differences between "younger" and "older" leaves can be justified.

Answer: Thank you so much for the detailed correction and suggestions. We have revised the manuscript according to your professional recommendations. Specificly, we have revised the figures in a reader-friendly manner, provided more detailed information about the figures. 

Reviewer 3 Report

Comments and Suggestions for Authors

The presented study by Guo Shiwei and co-authors entitled “Physiological Responses of Grapevine Leaves to High Temperature at Different Senescence Periods” is relevant and well-designed. The results provide insights for understanding heat-induced leaf senescence (including chlorophyll content and fluorescence parameters, MDA levels, and antioxidant enzyme activities) and developing strategies for cultivating grapevines. In general, the used methods are adequate and modern, and the obtained results are well described and discussed. The conclusions are clear, cover all the results obtained and outline further directions for research.

The paper may be accepted for publication after minor text editing.

Figures 2 and 6. It should be placed after the first mention in Figure 2 in the main text. The same comment for Figure 6.

Author Response

Q.1. The Figures 2 and 6. It should be placed after the first mention in Figure 2 in the main text. The same comment for Figure 6.

Answer: Thank you so much for your kind suggestions. We fully agree with you. We have revised the layout of the manuscript according to your professional recommendations.

Reviewer 4 Report

Comments and Suggestions for Authors

This study investigated the effect of high temperatures on the senescence traits of grapevine leaves, such as chlorophyll content, chlorophyll fluorescence, and antioxidant enzyme activity. Overall, the manuscript is well structured, the objective of the study is stated clearly and understandably, and the conclusions are supported by the results.
A few questions and comments.
1. The introduction section should be supplemented with a literature review of similar studies. The authors in this section only formulate the scientific problem and the objective of the study, without reviewing previous studies on this topic.
2. In the materials and methods section, the amount of precipitation and irrigation parameters, the number of irrigations should be indicated, since these parameters also affect leaf physiology. Were the temperatures measured in the root zone?
3. For figures 3-6, captions and legends should be added to each graph a, b c, etc. Also, the font size should be increased in figure 4, the axis labels and legends are almost illegible.
4. Lines 188-191 should probably be moved to section 4.3.3

Author Response

Q.1. The introduction section should be supplemented with a literature review of similar studies. The authors in this section only formulate the scientific problem and the objective of the study, without reviewing previous studies on this topic.

Answer: Thank you so much for your kind suggestions. We have added a short paragraph in the introduction to provide more information about the literature review of similar studies, and fully reviewed previous studies on this topic (Lines 45-60). Specificly, a concise mini-review (nine references including five newly added papers) was inserted immediately after the background statement. This section summarizes recent crop-level evidence (cucumber, cotton, tomato, maple and soybean) showing that leaf age/position critically determines heat or drought induced senescence kinetics, and that chlorophyll fluorescence and antioxidant traits are reliable indicators for quantifying age-specific responses. The added text explicitly states that "whether similar age-dependent patterns occur in field-grown grapevines repeatedly exposed to > 40 °C heatwaves remains largely unexplored", thus clearly identifying the knowledge gap addressed by the present study.

Q.2. In the materials and methods section, the amount of precipitation and irrigation parameters, and the number of irrigations should be indicated, since these parameters also affect leaf physiology. Were the temperatures measured in the root zone?

Answer: Thank you for considering the details on experimental procedures. We have added a short paragraph in 4.1 Description of the Study Area and 4.2 Plant Materials to provide more information about precipitation and irrigation(Lines 558-559, 566-569). Unfortuenately, we have not measured the temperatures in the root zone in this experionment, because we only focus on air temperature (nearly 1.0 meter high in the grapevine garden, almost half of plant’s high) in this reseearch.

Q.3.  For figures 3-6, captions and legends should be added to each graph a, b, c, etc. Also, the font size should be increased in Figure 4; the axis labels and legends are almost illegible.

Answer: Thank you so much for your kind suggestions. Caption and legends have been added to each figure, as you have mentioned. Particularly in Figure 4, we have revised the axis labels and legends carefully according to your recommendations.

Q.4. Lines 188-191 should probably be moved to section 4.3.3

Answer: Thank you for considering the details in the text and the suggestion. We have seriously considered moving them to section 4.3.3. Whereas we think it’s easier to understand these chlorophyll fluorescence parameters for readers in this way, particularly for some important parameters. In addition, readers might feel confused if they lose the key information about these indicators.